# Exploring the Role of Different Cell-Death-Related Genes in Sepsis Diagnosis Using a Machine Learning Algorithm

**DOI:** 10.3390/ijms241914720

**Published:** 2023-09-29

**Authors:** Xuesong Wang, Ziyi Wang, Zhe Guo, Ziwen Wang, Feng Chen, Zhong Wang

**Affiliations:** 1School of Clinical Medicine, Tsinghua University, Beijing 100190, China; wxs1103@outlook.com (X.W.); ziyi-wan21@mails.tsinghua.edu.cn (Z.W.); wang-zw20@mails.tsinghua.edu.cn (Z.W.); cfs01550@btch.edu.cn (F.C.); 2Beijing Tsinghua Changgung Hospital, Tsinghua University, Beijing 100084, China; gza01482@btch.edu.cn

**Keywords:** sepsis, programmed cell death, machine learning, immunity

## Abstract

Sepsis, a disease caused by severe infection, has a high mortality rate. At present, there is a lack of reliable algorithmic models for biomarker mining and diagnostic model construction for sepsis. Programmed cell death (PCD) has been shown to play a vital role in disease occurrence and progression, and different PCD-related genes have the potential to be targeted for the treatment of sepsis. In this paper, we analyzed PCD-related genes in sepsis. Implicated PCD processes include apoptosis, necroptosis, ferroptosis, pyroptosis, netotic cell death, entotic cell death, lysosome-dependent cell death, parthanatos, autophagy-dependent cell death, oxeiptosis, and alkaliptosis. We screened for diagnostic-related genes and constructed models for diagnosing sepsis using multiple machine-learning models. In addition, the immune landscape of sepsis was analyzed based on the diagnosis-related genes that were obtained. In this paper, 10 diagnosis-related genes were screened for using machine learning algorithms, and diagnostic models were constructed. The diagnostic model was validated in the internal and external test sets, and the Area Under Curve (AUC) reached 0.7951 in the internal test set and 0.9627 in the external test set. Furthermore, we verified the diagnostic gene via a qPCR experiment. The diagnostic-related genes and diagnostic genes obtained in this paper can be utilized as a reference for clinical sepsis diagnosis. The results of this study can act as a reference for the clinical diagnosis of sepsis and for target discovery for potential therapeutic drugs.

## 1. Introduction

Sepsis is a disease induced by a dysregulated immune response to infection, similar to infectious shock [1]. It is currently a public health burden worldwide due to its high morbidity and mortality [2,3]. According to the 3.0 definition of sepsis, the current diagnosis of sepsis mainly relies on sequential organ failure assessment score (SOFA) scores, which provide an accurate basis for a sepsis diagnosis. However, the diagnosis window is relatively late, and there is some delay in the early treatment of sepsis patients [4]. Konrad Reinhart et al. reviewed new approaches in molecular diagnosis and biomarker discovery in sepsis, and envisioned improved molecular diagnosis and prognosis based on transcriptome, proteome, or metabolic profiling to better understand the complexity of systemic inflammation [5]. Nowadays, many biomarkers (C-reactive protein, procalcitonin, etc.) have been evaluated for sepsis diagnosis and condition assessment [6]. However, the sensitivity and specificity of these markers in the diagnostic process may need to be improved. Machine learning algorithms have been widely adopted for biomarker probing and diagnostic model building. Li et al. identified the role of iron death-associated genes in the diagnosis of childhood sepsis and identified MAPK14 as a diagnostically relevant marker via logistic regression (LR) [7]. Zhang et al. integrated several genetic datasets of sepsis and identified two sepsis subtypes using deep learning and found significant differences in mortality between the two subtypes [8].

Programmed cell death (PCD), an essential form of cell death, has been proven to be closely related to the occurrence and development of many diseases. PCD specifically includes apoptosis, necroptosis, ferroptosis, pyroptosis, netotic cell death, entotic cell death, lysosome-dependent cell death, parthanatos, twelve forms of autophagy-dependent cell death, oxeiptosis, and alkaliptosis [9]. Apoptosis refers to the autonomous and orderly death of cells controlled by genes in order to maintain the stability of the internal environment [10]. Necroptosis refers to the process of self-destruction of cells activated by an extracellular signal (death receptor–ligand binding) or intracellular signal (foreign microbial nucleic acid) when apoptosis is blocked. It is closely related to a variety of inflammatory diseases and can cause the host to resist the invasion of viruses and microorganisms [11]. Unlike the above-mentioned PCD forms, iron death means that under the action of divalent iron or ester oxygenase, unsaturated fatty acids that are highly expressed on the cell membrane are catalyzed to undergo lipid peroxidation, thus inducing cell death [12]. Iron death is often accompanied by a large amount of iron accumulation and lipid peroxidation. Pyroptosis is the main mode of cell death that may be triggered after pathogen infection [13]. Insect disease is a mechanism of cell competition in cancer, and entotic cell death refers to cell death caused by phagocytosis and the killing of neighboring cells [14]. Lysosomal-dependent cell death is mediated by hydrolase (cathepsin) or iron released by Lysosomal Membrane Permeabilization (LMP), which is characterized by lysosomal rupture [15]. Parthanatos is a form of PARP1-dependent PCD, which is activated by DNA damage induced by oxidative stress [16]. Autophagy-dependent cell death is a kind of PCD driven by the molecular autophagy mechanism, which is characterized by autophagy vacuolation [17]. Oxeiptosis has an anti-inflammatory effect when activated by an increase in intracellular active oxygen levels and when encountering pathogens, and it also has a potential influence on pathogen clearance and teratogenic cells [18]. Alkaliptosis is a form of PCD regulated by intracellular alkalization [19].

Several previous studies have focused on the role of multiple PCD-related genes in the diagnosis of sepsis. Wang et al. used multiple machine learning algorithms to screen out eight genes (CLEC1A, MALT4, NAIP, NLRC1, SERPINB1, SIRT3, STAT2, and TLR0) related to sepsis diagnosis from a collection of apoptosis-related genes [20]. Liang et al. used univariate Cox analysis and least absolute shrinkage and selection operator (LASSO) Cox regression analysis to develop and validate a predictive risk score for sepsis prognosis based on six pyroptosis-related genes [21]. Hao et al. used the LASSO and support vector mechanism to build a new diagnostic model with six mitochondrial correlations [22].

This paper aims to deeply explore the relationship between PCD-related genes and sepsis through bioinformatics analysis and machine learning methods. Specifically, we analyzed the differential expression of PCD-related genes based on the microarray data of sepsis in the Gene Expression Omnibus (GEO) database (https://www.ncbi.nlm.nih.gov/geo/) accessed on 15 January 2023, and from a control group. Then, we obtained differential expression genes (DEGs). Based on various machine learning algorithms, we screened for diagnosis-related genes based on the gene expression data from the sepsis and control group and built a diagnosis model. In addition, the immune landscape of sepsis was discussed based on diagnosis-related genes.

## 2. Results

### 2.1. Differential Expression Analysis and Enrichment Analysis

Figure 1 shows the overall flowchart of this paper. In this paper, PCD-related genes were collected from a previous study [9], and the expression of these genes was extracted from the transcriptomic data species in the GSE26440 dataset. The differential expression analysis was then performed with the Limma algorithm, and DEGs were obtained. Heat maps of the expression of DEGs in the sepsis and control groups and volcano maps obtained from the differential expression analysis are presented in Figure 2A,B, respectively. Figure 2C provides the protein–protein interactions (PPI) network constructed from the DEGs. We identified the hub genes in the PPI network based on three algorithms: Closeness, EPC, and Radiality. In addition, we conducted Metascape enrichment analysis (Figure 3A) and Kyoto Encyclopedia of Genes and Genomes (KEGG) enrichment analysis (Figure 3B) on the DEGs. The Section 3 will analyze the relationship between the significant pathways and sepsis in detail.

### 2.2. Construction of the Diagnosis-Related Model Based on Machine Learning Algorithms

In this paper, we used the RF algorithm to analyze the weighted histogram of DEGs (Figure 4A). Then, we verified the diagnostic performance of genes in both the internal (samples in the test set of the GSE26440 dataset) and external test sets (all samples in the GSE95233 dataset). Specifically, the larger weight of a gene represents its greater importance in diagnosis. For this reason, we ranked genes by weight size and then retained 26 genes with weights greater than the mean. To identify the genes most relevant to the diagnosis of sepsis, the first, first two, etc., of the training set were then separately ranked. The expressions corresponding to the first 10 genes were sequentially entered into the LR, RF, SVM, DNN, Adaboost, and KNN models for training the model. We statistically calculated the change in the AUC of the internal test set during the process, as shown in Figure 4B. We tallied the ROC curves of several machine learning algorithms when the internal test set reached the maximum AUC, as shown in Figure 5A–E. When the external test set reached the maximum AUC, the ROC curves of several machine learning algorithms were calculated, as shown in Figure 5F–J. We found that the top 10 genes using the RF algorithm could achieve the maximum AUC in the internal test set. Furthermore, we verified whether the expression of these diagnostically relevant genes differed between sepsis and the control, and whether it was diagnostically significant for sepsis. The ROC curves of diagnostic-related genes in the internal and external test sets are given in Appendix A, respectively. Among them, the AUCs of ITGAM, KIF1B, RRAGD, S100A9, SCOC, and SPTLC2 in the internal and external test sets were more than 0.6, indicating diagnostic significance for sepsis.

### 2.3. Exploration of the Immune Landscape

In this paper, the expression of diagnosis-related genes in the GSE26440 dataset was input into the CIBERSORT algorithm, and then the infiltration abundance of immune cells in all samples was output. Figure 6A shows the boxplot of the difference in the infiltration abundance of immune cells between the sepsis group and the control group. As can be seen from the figure, there are significant differences between the two groups in the Tregs, M0 macrophages, M1 macrophages, activated mast cells, and neutrophils. Figure 6B is the difference boxplot of the HLA-related gene expression between the two groups. Among them, the expressions of HLA-C, HLA-DMA, HLA-DMB, HLA-DOA, HLA-DOB, HLA-DPA1, HLA-DPB1, HLA-DPB2, HLA-DRA, and HLA-DRB6 are significantly different between the two groups. Figure 6C gives the expression heat map of other immune cells in the sepsis and control group after removing the immune cells with an expression of 0 in more than half of the samples. Figure 6D–M show the correlation thermograms of diagnosis-related genes and HLA-related genes. In addition, a scatter plot of the correlation between diagnostic genes and immune cells was drawn based on the Pearson correlation coefficient (Figure 7 and the corresponding folder that can be found in the Appendix A). The gray shaded portion of each subplot represents the confidence interval for the regression line, i.e., the uncertainty of the estimated regression line. Its width depends on the dispersion of the data. According to the results of the qPCR experiment (Figure 8), we found that compared with the control group, the expression levels of the ITGAM (95% CI: −2.232~−0.900), KIF1B (95% CI: −2.034~−1.315), MMP9 (95% CI: −3.022~−0.962), RRAGD (95% CI: −1.394~−0.515), S100A9 (95% CI: −3.362~−1.099), and SORT1 (95% CI: −1.700~−0.864) genes in patients in the sepsis group were the most significant, followed by those of the SCOC (95% CI: 0.446~1.389), WDFY3 (95% CI: −2.239~−0.543), and SPTLC2 (95% CI: −1.190~−0.053) genes, and there was no significant difference found in the expression levels of the SH3GLB1 (95% CI: −0.938~0.465) gene. In addition, we show the scatter plots that show the first five diagnostic genes, the last five diagnostic genes, and immune cells with a significant correlation in Appendix A, respectively.

### 2.4. Comparison with Diagnostic Models Constructed in Other Studies

In this study, we identified ITGAM, KIF1B, MMP9, RRAGD, S100A9, SCOC, SH3GLB1, SORT1, SPTLC2, and WDFY3 as genes associated with a sepsis diagnosis. Based on these genes, a diagnostic model of sepsis was constructed using a variety of machine learning algorithms. In order to confirm the performance of the diagnostic model constructed with the mined genes in this paper, it was compared with the performance of diagnostic models constructed in several other studies. Specifically, to ensure the fairness of the contrast experiments, we uniformly used the logistic regression algorithm provided by the IBM SPSS Statistics 26 software to construct the diagnostic model in the external test set. The studies included in the comparison were [23,24,25]. As can be seen from Figure 9, the diagnostic model constructed in this paper has the highest AUC.

## 3. Discussion

Sepsis, a complex syndrome caused by infection, has a high morbidity and mortality. However, the role of PCD-related genes in the occurrence and development of sepsis is still unclear. Therefore, from the perspective of computational biology, this paper determined the role of PCD-related genes in sepsis diagnosis in detail by using various machine learning algorithms. Firstly, the expression of PCD-related genes in the GSE26440 dataset was extracted. Then, DEGs were obtained via differential expression analysis using the Limma algorithm. Through the Metascape and KEGG enrichment analysis of the DEGs, we found that many significant enriched pathways were closely related to the occurrence and development of sepsis. Sepsis is a systemic inflammatory response syndrome caused by infection, and its core mechanism is immune dysfunction. Endothelial dysfunction and neutrophil degranulation are the central events in the pathophysiology of sepsis [26]. The occurrence of sepsis is related to an imbalance in the inflammatory response [27]. Sepsis patients show a high inflammatory stage initially, which then evolves into a more lasting immunosuppression stage [28]. During sepsis, cell death caused by apoptosis will directly lead to microvascular dysfunction and organ failure in sepsis patients, and apoptotic immune cells will lead to secondary infection or immunosuppression [29]. Autophagy dysfunction leads to CD4^+^ T-cell apoptosis during sepsis. Wang et al. found that mTOR deletion can improve this symptom by improving autophagy–lysosomal fusion [30]. In a study of patients with acute myeloid leukemia (AML) and sepsis in Texas hospitals, the researchers found that the incidence of sepsis in AML patients was higher and the overall mortality was higher [31].

Furthermore, we used machine learning algorithms (LR, RF, SVM, DNN, Adaboost, and KNN) to screen for diagnosis-related genes (ITGAM, KIF1B, MMP9, RRAGD, S100A9, SCOC, SH3GLB1, SORT1, SPTLC2, and WDFY3). Zhou et al. indicated that the progress of sepsis could be caused by ITGAM, promoting nuclear and cytoplasmic translocation and the activated release of HMGB1 [32]. In a study analyzing sepsis and coronavirus disease 2019 (COVID-19) through bioinformatics and systems biology approaches, ITGAM was identified as a potential key biomarker based on regulatory analysis [33]. Zhang et al. explored sepsis-related co-expression modules through weighted gene co-expression network analysis (WGCNA). They identified ITGAM as a hub gene in the mRNA-lncRNA-Pathway co-expression network [34]. KIF1B encodes a motor protein that transports mitochondria and synaptic vesicle precursors [35]. In the early stage of sepsis, mitochondrial respiration is directly blocked by accumulated nitric oxide, which leads to body shock [36]. This indirectly indicates that KIF1B may play a role in septic shock. In a study aiming to identify sepsis in children based on machine learning, the authors identified KIF1B as a pivotal gene using LASSO and RF algorithms [37]. TFPI is an endogenous inhibitor of TF, and activated MMP-9 will reduce TFPI. Based on the single-cell transcriptome (scRNA-seq) data analysis of neutrophils, Hong et al. found that a subpopulation with high MMP9 expression is more accurate in predicting septic shock [38]. There is evidence that the lower level of TFPI is closely related to sepsis [39,40]. S100A9 is a calcium- and zinc-binding protein, which plays an important role in regulating the inflammatory process and immune response [41]. S100A9 may be a key target for the treatment of sepsis [42]. Dai et al. found that intracellular S100A9 promotes myeloid-derived suppressor cells in late sepsis [43]. Zhang et al. found that S100A9 plays an essential pro-inflammatory role in sepsis-mediated acute liver injury by regulating AKT-AMPK-dependent mitochondrial energy metabolism [44]. SH3GLB1 is a Bax-binding protein, which is involved in autophagy and apoptosis [45]. SORT1 may affect the prognosis of sepsis and patient survival [46]. Ciclopirox reduces inflammation through the SORT1-mediated Wnt/β-catenin signaling pathway, thereby preventing septic shock [47]. Di et al. identified WDFY3 as a hub gene in sepsis from autophagy-related genes through bioinformatics analysis [48].

We constructed a diagnosis model of sepsis by using diagnosis-related genes, and the AUC of the model in the internal and external test sets reached 0.7951 and 0.9627, respectively. Furthermore, we discussed the immune landscape of sepsis based on diagnosis-related genes. Among them, the immune infiltration abundance of Tregs, M0 macrophages, M1 macrophages, activated mast cells, and neutrophil cells was significantly different between the sepsis group and control group. These immune cells all play an important role in the occurrence and development of sepsis. Tregs are a subset of CD4^+^ T lymphocytes, and their function is to control the immune response. Sepsis can increase the heterogeneity of Tregs, and thus regulating the Tregs is an important strategy to improve the symptoms of sepsis [49,50]. Macrophages play a vital role in the important cells of the innate immune system. Macrophages have two phenotypes of activation, which are broadly called M1 and M2 polarization. The overwhelming inflammation induced by the M1 phenotype is closely related to sepsis [51], and the transformation from the M1 phenotype to the M2 phenotype can protect the body from excessive inflammatory damage. Studies have shown that autophagy can cause the polarization of macrophages [52], and macrophage autophagy can lead to macrophage death by promoting autophagy death or apoptosis in sepsis [53]. In addition, Zhang et al. found that the increase in M0 macrophages in peripheral blood is an independent risk factor for a poor prognosis of sepsis [54]. Mastercells are important immune cells derived from hematopoietic stem cells. Yue et al. found that neuroinflammation in septic mice is related to the activation of Mastercells [55]. Neutrophils are an important part of the innate immune response, and neutrophil migration will change during the many stages of sepsis, which will eventually lead to innate immune deficiency in sepsis patients [56].

In addition, through the analysis of the expression of HLA-related genes in the two groups, we found that there were significant differences in the expression of HLA-C, HLA-DMA, HLA-DMB, HA-DOA, HLA-DOB, HLA-DPA1, HLA-DPB1, HLA-DPB2, HLA-DRA, and HLA-DRB6 in the sepsis and control groups. The human leukocyte antigen (HLA) complex plays an important role in immune system functioning. The expression of PDE4D in sepsis patients decreased continuously with time, and the low expression level of PDE4D was found to be related to HLA-DMA and HLA-DMB [57]. Mohsin et al. used comprehensive transcriptomics and regulatory network analysis methods to reveal the role of HLA-DPB1 in sepsis [58]. Experiments have confirmed that monitoring HLA-DRA by using qRT-PCR can help us to detect the dynamic changes in the immune state in sepsis, which is beneficial to future research [59].

## 4. Materials and Methods

### 4.1. Data Acquisition

In this paper, two sets of sepsis transcriptome data (GSE26440 and GSE95233) were downloaded from the GEO database. We used GSE26440 as an internal training set and test set, which included 98 sepsis samples and 32 control samples. GSE95233 was applied as the test set, which included 51 sepsis samples and 22 control samples.

### 4.2. Differential Expression Analysis

We collected PCD-related genes from a previous study [9]. The expression levels of these genes were extracted from the transcriptome data of the GSE26440 dataset. The expression matrix and its sample labels were used as input, and then the Limma algorithm [60] was used to analyze the differential expression. The Limma algorithm was implemented in this paper based on the R package “limma”. *p* < 0.05 and |logFC| > 1 were set during analysis. Finally, 59 DEGs were output.

### 4.3. Protein Interaction Network Construction

In this paper, we imported the DEGs into the String database (http://string.embl.de/) accessed on 25 January 2023, and exported the protein–protein interaction (PPI) network construction results. Then, we visualized the network with Cytoscape software (v3.9.2). Based on the three algorithms of Closeness, EPC, and Radiality from the cytoHubba plugin of the Cytoscape software, we identified the hub genes in the PPI network.

### 4.4. Enrichment Analysis

The Metascape database (http://metascape.org/) accessed on 24 September 2023, is a powerful tool for the functional annotation analysis of genes. We took the names of the DEGs as input to obtain the final results of the enrichment analysis. In addition, we acquired the results of the (KEGG) enrichment analysis through the R package “clusterProfiler.” For both enrichment analyses, we retained a significant pathway of *p* < 0.05.

### 4.5. Selection of Diagnostic-Related Genes and Construction of Diagnostic Models

In this study, we selected feature genes based on a Random Forest (RF) algorithm and provided the weight information of each gene. Then, the features were ranked by the weights, and the features with higher weights were considered to be more important. We implemented five machine learning algorithms: an RF, support vector machine (SVM), deep neural network (DNN), k-nearest neighbor (KNN), and Adaboost, which is based on the scikit-learn package for Python. The training strategy was to divide the GSE26440 dataset into a training set and a test set. The training set included 80 sepsis samples and 24 control group samples, and the test set included 18 sepsis samples and 8 control group samples. Then, we picked the best parameters for each classifier using a five-fold cross-validation method for the training set samples, and then trained the model based on the best parameters and tested the model on the internal and external test sets. In addition, we set a fixed random seed for the RF, SVM, DNN, KNN, and Adaboost classifiers and saved the training models as several other classifiers.

### 4.6. Immune-Related Analysis

In this study, we explored the differences between sepsis and control samples in terms of immune cell infiltration and HLA-related genes with diagnostic-related genes. Specifically, this paper estimated the abundance of 22 immune cell infiltrates in sepsis and control samples based on diagnosis-related genes using the CIBERSORT algorithm. In addition, HLA-associated genes were collected in this work.

### 4.7. qPCR Experimental Verification

In order to verify that 10 diagnosis-related genes were screened out by the machine learning algorithm, this paper used 8 samples from sepsis patients (sepsis group) and 8 samples from normal people (control group) to conduct real-time quantitative PCR experiments. Demographic information for the sepsis group and the control group is shown in Appendix A. The screening criteria for sepsis patients were based on sepsis 3.0 (1). This study was approved by the Ethics Committee (NCT0509324).

## 5. Conclusions

In summary, this paper screened for genes related to sepsis diagnosis using machine learning methods, and built a diagnostic model based on the findings. In addition, the immune landscape of these genes was analyzed in detail and verified experimentally. The results of this study can act as a reference for the clinical diagnosis of sepsis and target discovery of potential therapeutic drugs.

Additionally, this paper has some limitations. Limited by the number of images in the sample, the machine learning model constructed in this article may be significantly affected by data selection bias. This results in limited generalization performance of the model. Since transcriptome data are usually collected at different time points or conditions, this may cause the expression of hub genes to change under different conditions. The GSE95233 dataset used in this article consists of patients with septic shock who were sampled twice on admission, with the second sample taken on the second or third day. That is, each sample contains data from at least two time points. Therefore, the GSE95233 dataset can be applied to mining pathways and core genes related to the progression of sepsis over time. Specifically, time-constrained coefficient learning [61] or a classic deep learning model (suitable for datasets with large sample sizes and many time points) can be applied to model time series data [62].The attention mechanism can also be introduced to explore the changes in the importance of disease-related biomarkers over time, providing a reference for drug target discovery. Therefore, future studies can collect transcriptome data at different time points in sepsis patients, and mine pathways and hub genes related to the progression of sepsis over time.

## Figures and Tables

**Figure 1 ijms-24-14720-f001:**
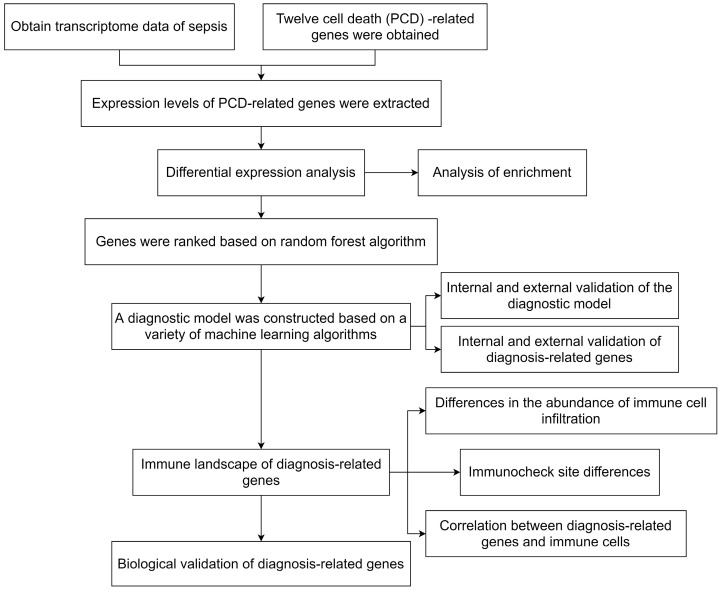
The overall flowchart.

**Figure 2 ijms-24-14720-f002:**
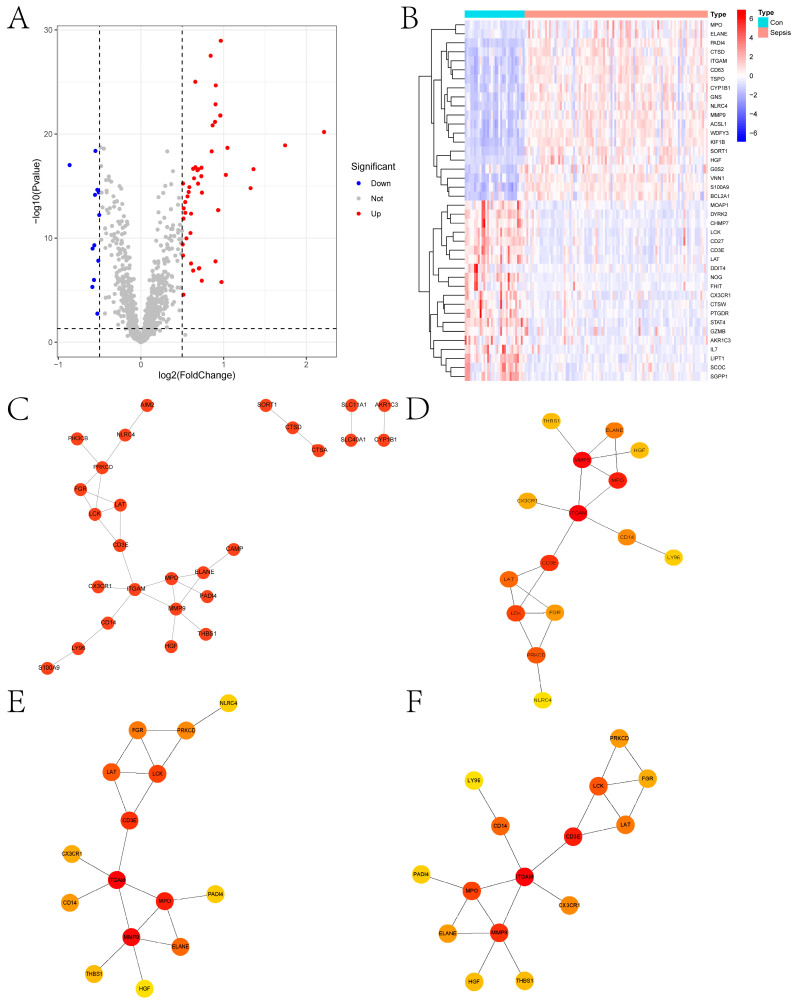
Differential expression analysis and PPI analysis. (**A**,**B**) show the heat map and volcano map obtained from differential expression analysis, respectively. (**C**) shows the PPI network constructed using DEGs. (**D**–**F**) show the identification of hub genes in the PPI network based on the three algorithms of Closeness, EPC, and Radiality, respectively.

**Figure 3 ijms-24-14720-f003:**
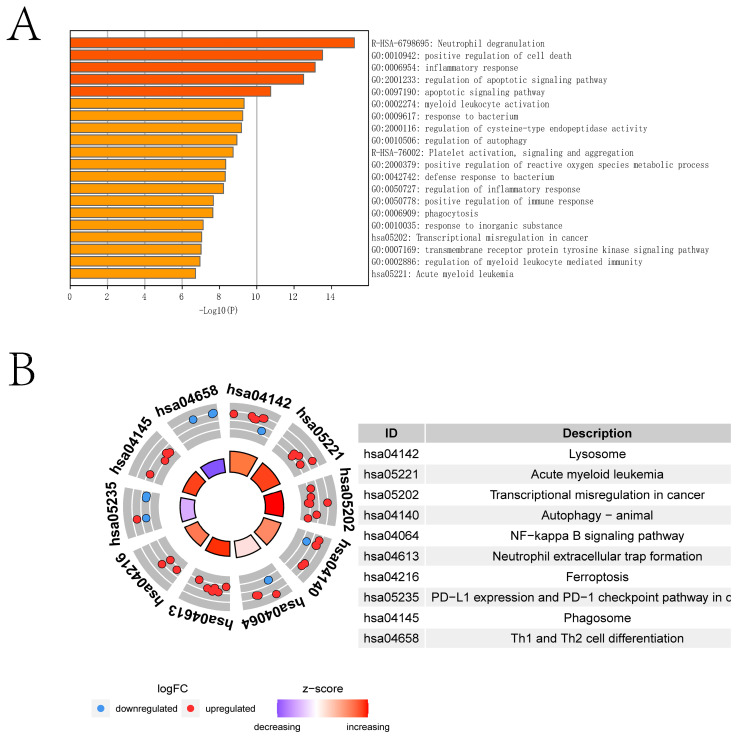
Enrichment analysis of DEGs. (**A**,**B**) show the results of Metascape and KEGG enrichment analysis of DEGs, respectively.

**Figure 4 ijms-24-14720-f004:**
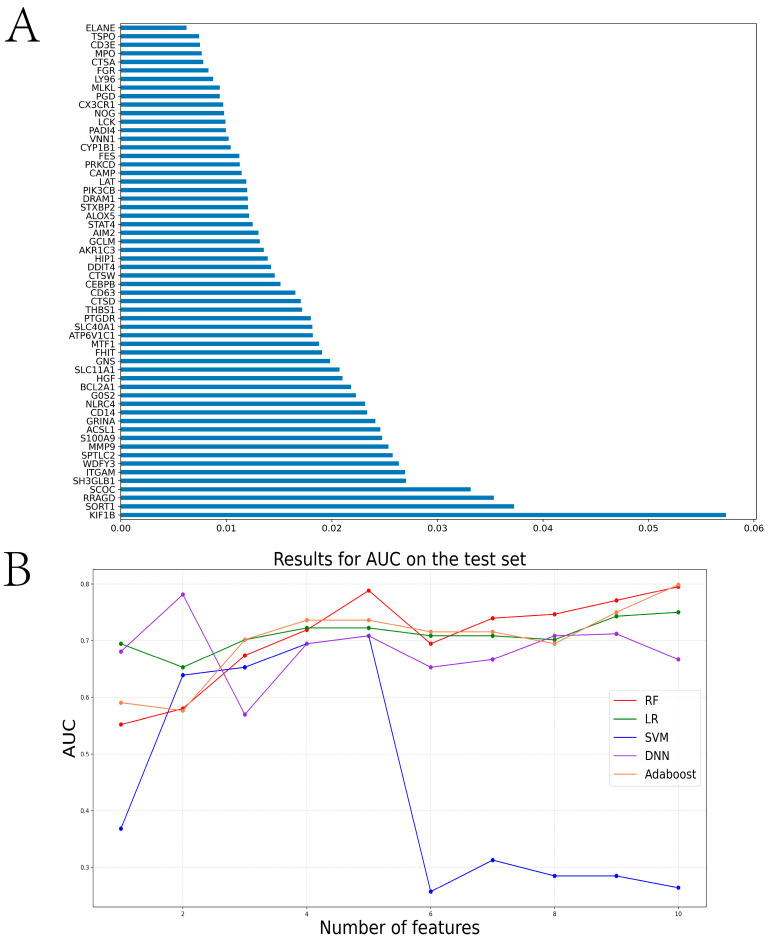
Twenty-six diagnosis- and PCD-related genes obtained via screening based on the RF algorithm. (**A**) shows the histogram of the weights of the 26 genes obtained by using the RF algorithm. (**B**) shows the line graph of the AUC validation of several machine learning algorithms on the diagnostic model in the internal test set after selecting different numbers of genes.

**Figure 5 ijms-24-14720-f005:**
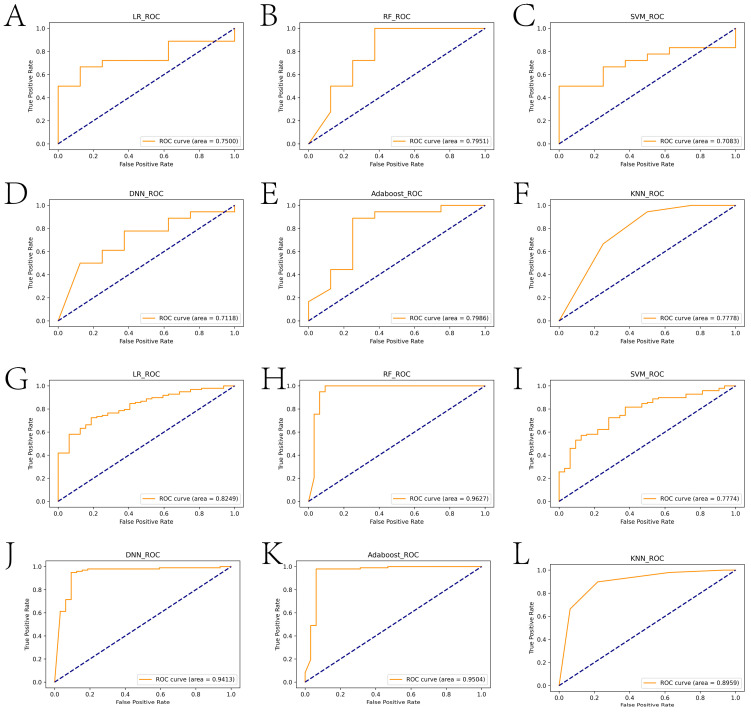
ROC curves of different machine learning algorithms when reaching the maximum AUC in the internal and external test sets. The blue diagonal typically represents the random guess case. The yellow line represents the relationship between True Positive Rate and False Positive Rate at different thresholds. (**A**–**F**) show the ROC curves of LR, RF, SVM, DNN, Adaboost, and KNN in the internal test set, respectively. (**G**–**L**) show the ROC curves of LR, RF, SVM, DNN, Adaboost, and KNN in the external test set, respectively.

**Figure 6 ijms-24-14720-f006:**
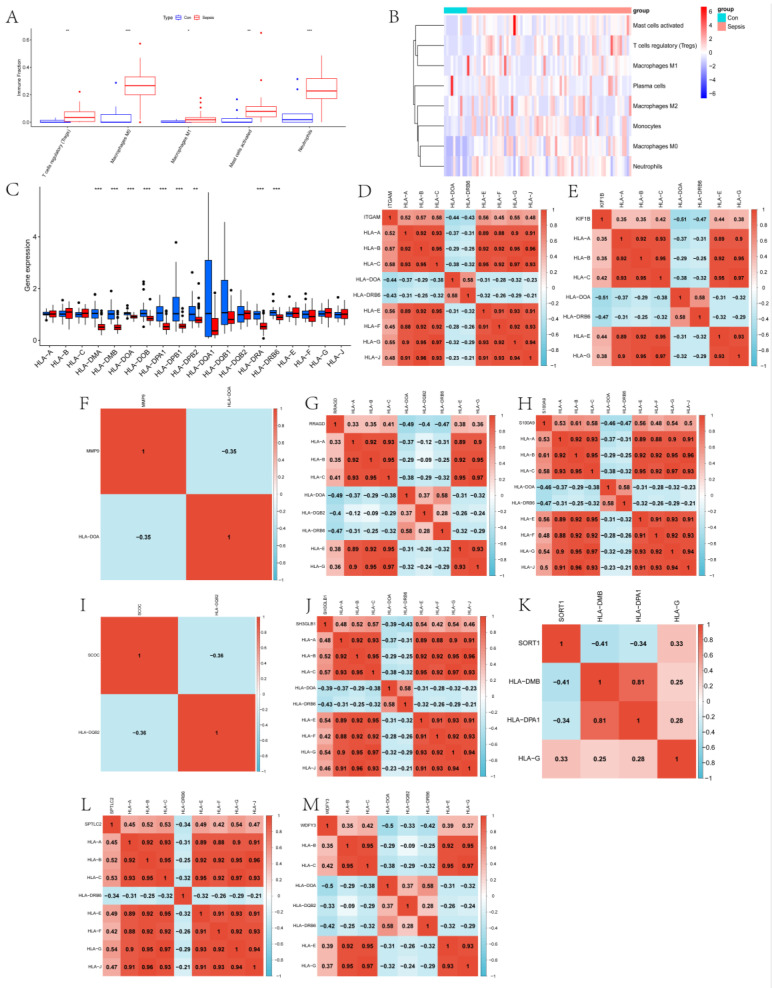
Immune landscape analysis of the sepsis and control groups. (**A**) shows the box diagram of the difference in immune cell infiltration abundance between the two groups. (**B**) shows the difference boxplot of the difference in expression of HLA-related genes between the two groups. (**C**) shows the expression heat map of immune cells in the sepsis and control groups. (**D**–**M**) show the correlation thermograms of ITGAM-, KIF1B-, MMP9-, RRAGD-, S100A9-, SCOC-, SH3GLB1-, SORT1-, SPTLC2-, WDFY3-, and HLA-related genes, respectively. Notes: ITGAM: Integrin Subunit α M; KIF1B: Kinesin Family Member 1B; MMP9: Matrix Metalloprotease 9; RRAGD: Ras related GTP binding D; S100A9: S100 calcium-binding protein A9; SCOC: Short coiled-coil protein; SH3GLB1: SH3 domain GRB2-like endophilin B1; SORT1: Sortilin1; SPTLC2: Serine palmitoyltransferase long chain base subunit 2; WDFY3: Autophagy-linked FYVE protein; HLA: Human leukocyte antigen. *** *p* < 0.001, ** *p* < 0.01, * *p* < 0.05.

**Figure 7 ijms-24-14720-f007:**
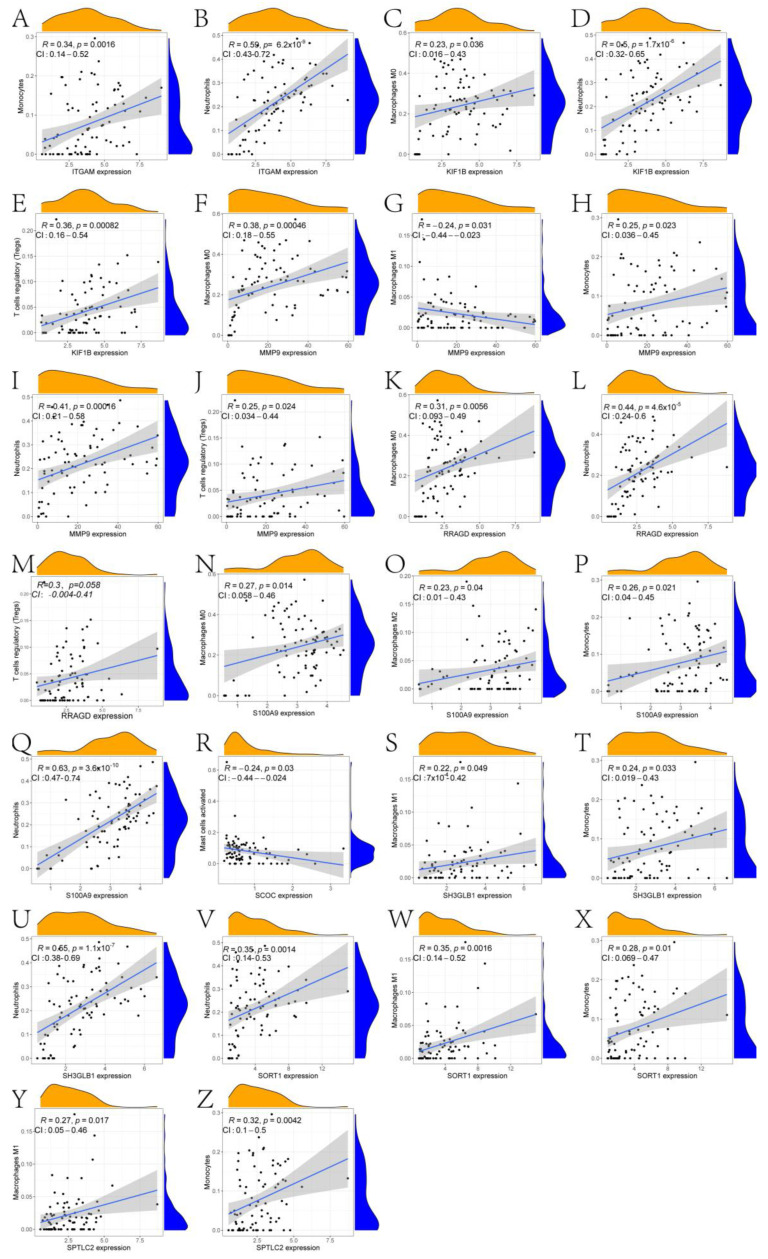
Scatter diagram of correlation analysis between the first five diagnosis-related genes and immune cells. (**A**,**B**) are correlation scatter plots between ITGAM and immune cells (Monocytes and Neutrophils). (**C**–**E**) are correlation scatter plots of KIF1B and immune cells (Macrophages M0, Neutrophils and T cells regulatory (Tregs)). (**F**–**J**) are correlation scatter plots of MMP9 and immune cells with significant correlation (Macrophages M0, Macrophages M1, Monocytes, Neutrophils and T cells regulatory (Tregs)). (**K**–**M**) is the correlation scatter plot of RRAGD and immune cells (Macrophages M0, Neutrophils and T cells regulatory (Tregs)). (**N**–**Q**) is the correlation scatter plot between S100A9 and immune cells (Macrophages M0, Macrophages M2, Monocytes and Neutrophils) that are significantly correlated with it. (**R**) is a correlation scatter plot between SCOC and immune cells (Mast cells activated). (**S**–**U**) is a correlation scatter plot of SH3GLB1 and immune cells (Macrophages M1, Monocytes and Neutrophils). (**V**–**X**) is a correlation scatter plot between SORT1 and immune cells (Macrophages M1, Monocytes and Neutrophils). (**Y**,**Z**) are correlation scatter plots between SPTLC2 and immune cells (Macrophages M1 and Monocytes). Notes: ITGAM: Integrin Subunit α M; KIF1B: Kinesin Family Member 1B; MMP9: Matrix Metalloprotease 9; RRAGD: Ras related GTP binding D; S100A9: S100 calcium-binding protein A9; SCOC: Short coiled-coil protein; SH3GLB1: SH3 domain GRB2-like endophilin B1; SORT1: Sortilin1; SPTLC2: Serine palmitoyltransferase long chain base subunit 2.

**Figure 8 ijms-24-14720-f008:**
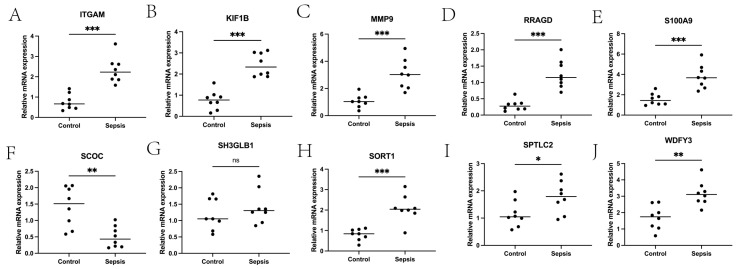
The qPCR verification results of diagnostic genes. (**A**–**J**) show box plots of the expression levels of ITGAM, KIF1B, MMP9, RRAGD, S100A9, SCOC, SH3GLB1, SORT1, SPTLC2, and WDFY3 in the control group and sepsis group, respectively. *** *p* < 0.001, ** *p* < 0.01, * *p* < 0.05.

**Figure 9 ijms-24-14720-f009:**
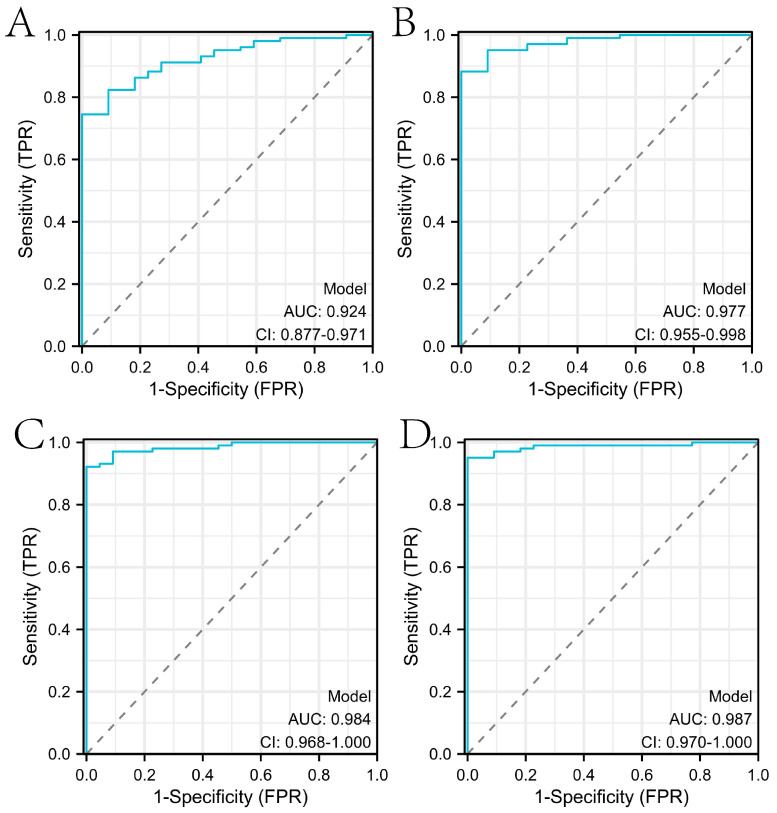
Results of comparison of diagnostic models. (**A**–**D**) show the ROC curves of the diagnostic models constructed [23,24,25], and the diagnostic genes mined in this paper, respectively.

## Data Availability

The original contributions presented in this study are included in the article/Appendix A. This study’s machine learning model code is available at https://github.com/wxs1103/ML-main accessed on 24 September 2023. Further inquiries can be directed to the corresponding author.

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
