# Peer review of "Exploring the Role of Different Cell-Death-Related Genes in Sepsis Diagnosis Using a Machine Learning Algorithm"

_ijms, 2023, doi:10.3390/ijms241914720_

Round 1
Reviewer 1 Report
This article sought to assess the role of different cell-death related genes in the context of sepsis diagnosis using a machine learning algorithm.
Overall, the manuscript was rather clearly presented.
In general, the manuscript’s English can be improved.
For example, line 35, “According to the definition of sepsis 3.0, the current diagnosis of sepsis mainly relies on SOFA scores, which provide accurate basis for the diagnosis of sepsis patients.” should be “According to the definition of sepsis 3.0, the current diagnosis of sepsis mainly relies on SOFA scores, which provide an accurate basis for a sepsis diagnosis.”
As another example, line 336, ” In a word, this paper screened genes related to sepsis diagnosis by machine learning method and built a diagnosis model based on this.” Should be “In summary, this paper screened genes related to sepsis diagnosis using machine learning methods and built a diagnostic model based on this.”
In addition, the manuscript has some formatting errors. For example, lines 82 through 84 should be removed.
In the methods, you describe well the different ML algorithms used. However, can you elaborate on why you selected these ML algorithms in particular, beyond the fact that they were based on the sciknit-learn package for Python? This is a methodological crux and one of the most salient aspects of your paper, of relevance to many genetics/ML researchers who will look to employ your methods and apply them to different diagnostic/clinical contexts.
Your figures are detailed but some of the panels are barely legible. Can you make Figure 2, panels A and B two times larger? Can you do the same for Figure 6, panels A through C? There is important information in these figures that is lost with the pixilation reduction.
In the conclusion, you mention that these results can provide reference for the clinical diagnosis of sepsis. It would add clout to your work to mention the important role of gene panels in the clinic, both in the context of prevention and treatment of various diseases, whether genetic in origin or with strong genetic signatures. How will your work advance this clinical implementation? How will it help patients and health care systems as a whole?
As above, this needs improvement throughout.
Author Response
Dear Editors and Reviewers:
Thanks for your letter and for the reviewers’ comments concerning our manuscript entitled “Explore the role of different cell death-related genes in sepsis diagnosis based on machine learning algorithm” (Manuscript ID:2561747). Those comments are all valuable and very helpful for revising and improving our paper, as well as the important guiding significance to our researches. We have studied comments carefully and have made correction, which we hope meet with approval. In addition, regarding to the reviewers’ insightful comments, the answers were listed one-by-one as follows and the correspondingly revised information marked in red could be found at the revised manuscript. Thanks again.
Comments and Suggestions for Authors
This article sought to assess the role of different cell-death related genes in the context of sepsis diagnosis using a machine learning algorithm. Overall, the manuscript was rather clearly presented.
Response: Thanks for the reviewer’s suggestions. We have carefully revised the paper based on your comments. This has greatly helped to improve the quality of the paper.
- In general, the manuscript’s English can be improved. For example, line 35, “According to the definition of sepsis 3.0, the current diagnosis of sepsis mainly relies on SOFA scores, which provide accurate basis for the diagnosis of sepsis patients.” should be “According to the definition of sepsis 3.0, the current diagnosis of sepsis mainly relies on SOFA scores, which provide an accurate basis for a sepsis diagnosis.” As another example, line 336, ” In a word, this paper screened genes related to sepsis diagnosis by machine learning method and built a diagnosis model based on this.” Should be “In summary, this paper screened genes related to sepsis diagnosis using machine learning methods and built a diagnostic model based on this.”
Response: Thanks for the reviewer’s suggestions. We have corrected several of the language issues you pointed out. We asked a professional team of native English speakers to polish this article.
- In addition, the manuscript has some formatting errors. For example, lines 82 through 84 should be removed.
Response: Thanks for the reviewer’s suggestions. We have deleted L82-L84. We also proofread the other parts of the content one by one and highlight them in red to ensure the quality of the paper.
- In the methods, you describe well the different ML algorithms used. However, can you elaborate on why you selected these ML algorithms in particular, beyond the fact that they were based on the sciknit-learn package for Python? This is a methodological crux and one of the most salient aspects of your paper, of relevance to many genetics/ML researchers who will look to employ your methods and apply them to different diagnostic/clinical contexts.
Response: Thanks for the reviewer’s suggestions. The machine learning algorithm chosen in this paper has advantages in different aspects. For example, the logistic regression algorithm is suitable for data with a small number of features, and the relationship between the features can be represented by a linear combination. However, random forest is an ensemble learning algorithm that can capture nonlinear relationships because it builds complex decision boundaries by combining multiple decision trees. It can also assess the importance of genes in diagnosis. To the best of our knowledge, the multiple machine learning models used in this paper are robust and reliable with low model complexity. This facilitates rapid use by genetics /ML researchers. In addition, through the comparison between multiple algorithms, the genes related to diagnosis can be better screened and the diagnostic model with high AUC can be constructed. We uploaded in making the model code (https://github.com/wxs1103/ML), which can be convenient researchers in other related data set for further research. Code links are supplemented to the abstract section.
- Your figures are detailed but some of the panels are barely legible. Can you make Figure 2, panels A and B two times larger? Can you do the same for Figure 6, panels A through C? There is important information in these figures that is lost with the pixilation reduction.
Response: Thanks for the reviewer’s suggestions. We have resized the neutron plots in Figures 2 and 6 to make them easy for readers to view. In addition, we uploaded PDF versions of Figures 1-7 separately.
- In the conclusion, you mention that these results can provide reference for the clinical diagnosis of sepsis. It would add clout to your work to mention the important role of gene panels in the clinic, both in the context of prevention and treatment of various diseases, whether genetic in origin or with strong genetic signatures. How will your work advance this clinical implementation? How will it help patients and health care systems as a whole?
Response: Thanks for the reviewer’s suggestions. In this study, we validated the differences in diagnostic related genes between sepsis and normal samples through qPCR, further demonstrating the important role of diagnostic related genes in the progression of sepsis. Diagnostic related genes may provide new targets for drug development and contribute to the development of sepsis related therapeutic drugs. In addition, monitoring the expression of diagnostic related genes helps to understand the pathological progression of sepsis patients. In summary, the sepsis related diagnostic genes identified in this study may provide new insights into the pathogenesis and progression of sepsis, and contribute to the treatment and prognosis of sepsis patients.
Reviewer 2 Report
Review on “Explore the role of different cell death-related genes in sepsis diagnosis based on machine learning algorithm” for IJMS (manuscript ID ijms-2561747)
The abstract structure differs from IJMS common practice and shouldn’t be divided into sections. Please avoid repetition in the abstract (L21 and L26).
In this manuscript the authors present an effort to discover the PCD-related genes with sepsis development using machine learning approach.
Comments on Introduction section:
Most of the Intro (L47-73) devoted to the PCD, not the related genes or previous studies on the topic. I strongly suggest to review the papers on PCD-related genes and mathematical models that can predict sepsis using gene expression data.
L42: the references [6-7] aren’t related with sepsis.
Please explain the choice of particular dataset (GSE26440).
L36: what “SOFA score” stands for?
L61: “Coke death”, what does it mean?
L17-18: Long sentence and starts with numbers, which make it complicated.
L71: which specific data used for screening?
My questions about Results and Discussion:
L82-84: please remove the text from template
L86: Figure 1 must be placed closer to referring text.
L183, L204: if authors describe 12 genes in the Intro section, why only 10 genes are presented in Results’ figures (Fig 3B, Fig 6)?
The scattering plots tells almost nothing (Fig 7) – the confidence interval must be too wide to define any relationship; please carefully evaluate the correlation coefficient (f.e. Pearson). Figure 7 legend must contain the information about the confidence interval and significance.
Pleas add the confidence interval description for the Figure 8.
Please compare the results with similar studies. In the present version authors mostly re-tell the results and point the perspective of the study.
Methods section comments:
· L256: The dataset GSE95233 is not mentioned in Results section.
· L283: Was the dataset GSE95233 used for training/testing?
· L261, L93: What is the Limma algorithm? Please provide reference and algorithm options if any. Please explain the choice of such algorithm.
· L87, L259: please cite the “previous literature”
· What is the purpose to use multiple ML algorithms without explaining the choice? Figure 5 not even mentioned in the main text! Is the trained model available for the further research? Please provide the source files to reproduce the results (via Github, for example).
· Description of each algorithm might be omitted (subsections 4.5.1–4.5.5)
Some minor corrections to the text (style and spelling):
· L50: replace comma to stop.
Author Response
Dear Editors and Reviewers:
Thanks for your letter and for the reviewers’ comments concerning our manuscript entitled “Explore the role of different cell death-related genes in sepsis diagnosis based on machine learning algorithm” (Manuscript ID:2561747). Those comments are all valuable and very helpful for revising and improving our paper, as well as the important guiding significance to our researches. We have studied comments carefully and have made correction, which we hope meet with approval. In addition, regarding to the reviewers’ insightful comments, the answers were listed one-by-one as follows and the correspondingly revised information marked in red could be found at the revised manuscript. Thanks again.
- The abstract structure differs from IJMS common practice and shouldn’t be divided into sections.
Response: Thanks for the reviewer’s suggestions. We have modified the structured presentation of abstracts to meet IJMS requirements.
- Please avoid repetition in the abstract (L21 and L26).
Response: Thanks for the reviewer’s suggestions. We have removed the repeated representations of L26.
- Most of the Intro (L47-73) devoted to the PCD, not the related genes or previous studies on the topic. I strongly suggest to review the papers on PCD-related genes and mathematical models that can predict sepsis using gene expression data.
Response: Thanks for the reviewer’s suggestions. We searched pubmed (pairs had been added after L73) for papers on 12 PCD-related genes, sepsis, and diagnosis. The relevant papers published in authoritative journals in the past five years were reviewed. This part is supplemented after L73.
- L42: the references [6-7] aren’t related with sepsis.
Response: Thanks for the reviewer’s suggestions. We have removed literature 6 and 7 not related to sepsis
- Please explain the choice of particular dataset (GSE26440).
Response: Thanks for the reviewer’s suggestions. In this study, we aim to explore the function and diagnostic significance of PCD related genes in sepsis. Machine learning algorithms require a sufficient sample size to make the selected diagnostic related genes more representative. Therefore, after reviewing sepsis related literature and sepsis related data in the GEO database, we ultimately chose the GSE26440 dataset.
- L36: what “SOFA score” stands for?
Response: Thanks for the reviewer’s suggestions. The SOFA score is mainly used to evaluate the level of organ dysfunction in patients. The diagnosis of sepsis 3.0 requires ΔSOFA≥2, but once patients experience organ dysfunction, their prognosis is often poor. Introducing SOFA score to indicate that there is still room for improvement in the current diagnostic methods for sepsis.
- L61: “Coke death”, what does it mean?
Response: Thanks for the reviewer’s suggestions. We have changed "Coke death" to "Pyroptosis" for L61.
- L17-18: Long sentence and starts with numbers, which make it complicated.
Response: Thanks for the reviewer’s suggestions. We have rewritten this long sentence to avoid starting with a number to make it easier for readers to understand.
- L71: which specific data used for screening?
Response: Thanks for the reviewer’s suggestions. We filtered based on gene expression data from sepsis and its control group (L85-86).
- L82-84: please remove the text from template
Response: Thanks for the reviewer’s suggestions. We have removed the template content.
- L86: Figure 1 must be placed closer to referring text.
Response: Thanks for the reviewer’s suggestions. We have moved Figure 1 to Section 2.1.
- L183, L204: if authors describe 12 genes in the Intro section, why only 10 genes are presented in Results’ figures (Fig 3B, Fig 6)?
Response: Thanks for the reviewer’s suggestions. The misunderstanding bias was caused by our unclear expression in the text. "12 PCD-related genes" refers to all the genes involved in the 12 PCD mentioned in this paper. We have removed the "12".
- The scattering plots tells almost nothing (Fig 7) – the confidence interval must be too wide to define any relationship; please carefully evaluate the correlation coefficient (f.e. Pearson). Figure 7 legend must contain the information about the confidence interval and significance.
Response: Thanks for the reviewer’s suggestions. We recalculated the correlation between immune cells and diagnostic genes using the Pearson method, and labeled the correlation value (R), significance value (p), and confidence interval (CI) in the upper left corner of the graph. The relevant content can be viewed in Figure 7 of the revised manuscript.
- Please add the confidence interval description for the Figure 8.
Response: Thanks for the reviewer’s suggestions. We have add the confidence interval description. (Line:175-181)
- Please compare the results with similar studies. In the present version authors mostly re-tell the results and point the perspective of the study.
Response: Thanks for the reviewer’s suggestions. We compared the diagnostic models constructed by the diagnostic genes found in several literatures with those constructed in this paper. Specifically, in order to ensure the rationality of the comparison, we built a diagnostic model based on the diagnosis related genes and the logistic regression algorithm for the literature and the independent test set GSE95233. The results show that the diagnosis model constructed by gene mining in this paper has higher AUC. This section has been added to the new section 2.4 in the main text.
- L256: The dataset GSE95233 is not mentioned in Results section.
Response: Thanks for the reviewer’s suggestions. Due to our oversight, only the two datasets used are briefly presented in the Methods section. The specific data sets used are given in our correlation analyses that address the use of data sets in the Results section (marked in red in the Results section). The GSE26440 dataset was used as the training set and internal test set. GSE95233 was used as an external test set.
- L283: Was the dataset GSE95233 used for training/testing?
Response: Thanks for the reviewer’s suggestions. We have supplemented the information on the use of this dataset in the results section and annotated in red.
- L261, L93: What is the Limma algorithm? Please provide reference and algorithm options if any. Please explain the choice of such algorithm.
Response: Thanks for the reviewer’s suggestions. We supplement the references of the Limma algorithm and the way it is implemented in this paper in Section 4.2.
- L87, L259: please cite the “previous literature”
Response: Thanks for the reviewer’s suggestions. We have supplemented the references to specific references in Sections 2.1 and 4.2.
- What is the purpose to use multiple ML algorithms without explaining the choice? Figure 5 not even mentioned in the main text! Is the trained model available for the further research? Please provide the source files to reproduce the results (via Github, for example).
Response: Thanks for the reviewer’s suggestions. To the best of our knowledge, the use of multiple machine learning algorithms to build diagnostic models can explore the data from different perspectives and methods, leading to more comprehensive and integrated analysis results. Different algorithms may have different ways to interpret and process the characteristics and patterns of the data. By combining multiple algorithms, bias can be reduced and the robustness of the model can be improved. Moreover, different machine learning algorithms have different advantages in dealing with different types of data and problems. The performance of different algorithms on independent test sets was compared to determine which algorithm performed best on sepsis transcriptome data. Therefore, the most popular and operable machine learning models were selected in this paper. In addition, the trained model has the ability of generalization and can effectively predict on independent data sets. We have uploaded the code to github. Code links are supplemented to the abstract section.
- Description of each algorithm might be omitted (subsections 4.5.1–4.5.5)
Response: Thanks for the reviewer’s suggestions. We have removed sections 4.5.1-4.5.5 from the main text.
- L50: replace comma to stop.
Response: Thanks for the reviewer’s suggestions. We have replaced the last comma of the sentence with "and".
Reviewer 3 Report
The authors have presented a novel method for identifying and screening genes for target discovery. They utilized machine learning algorithms to explore the relationship between PCD-related genes and sepsis and achieve their goal. The article has a strong scientific background and the results support their conclusions.
The article could be improved if the authors included more information on the source of the genes that were validated in this paper. Did the authors find any GWAS or MR studies that looked for this association?
Adding a limitations section is extremely important, especially as the majority of the work presented is computational and theoretical.
Author Response
Dear Editors and Reviewers:
Thanks for your letter and for the reviewer’s comments concerning our manuscript entitled “Explore the role of different cell death-related genes in sepsis diagnosis based on machine learning algorithm” (ijms-2561747). Those comments are all valuable and very helpful for revising and improving our paper, as well as the important guiding significance to our researches. We have studied comments carefully and have made correction, which we hope meet with approval. In addition, regarding to the reviewers’ insightful comments, the answers were listed as follows and the correspondingly revised information marked in red could be found at the revised manuscript. Thanks again.
Reviewer 3:
The authors have presented a novel method for identifying and screening genes for target discovery. They utilized machine learning algorithms to explore the relationship between PCD-related genes and sepsis and achieve their goal. The article has a strong scientific background and the results support their conclusions.
The article could be improved if the authors included more information on the source of the genes that were validated in this paper. Did the authors find any GWAS or MR studies that looked for this association?
Adding a limitations section is extremely important, especially as the majority of the work presented is computational and theoretical.
Response: Thanks for the reviewer’s suggestions. We used the keywords "(Mendelian randomization) AND (Sepsis) AND (hub gene)", "(GWAS) AND (Sepsis) AND (hub gene)" in PubMed (https://pubmed.ncbi.nlm.nih.gov/) gene)". We retrieved relevant literature. Regrettably, no relevant studies were found. However, more literature on hub genes in sepsis was discovered through “(Sepsis) AND (hub gene)”, and we added this part to the second paragraph of the Discussion section. In addition, information about the limitations of our study of this article has been added to the conclusion section to help readers understand the credibility and applicability of the study.

Reviewer 4 Report
This study is considered worthy of publication in this journal because it discovered 10 genes that can be used to determine sepsis by reanalyzing the publicly available transcriptome. However, improvements need to be made in the following points.
1. All sepsis authors should discuss the transcriptome material and sepsis information they are referring to. Sepsis has a process from establishment to severity, and it is necessary to discuss the period from the initial stage to severity of the referenced transcriptome by referring to the transcriptome-reported author's papers.
2 Regarding Figure 3, information on creating the diagram, such as the algorithm, should be added to the caption.
3 Regarding Figure 7, an explanation should be added regarding A-Z.
4. Explanation should be added regarding A-J in Figure 8 as well.
5 Regarding Figure 9, Chen, Lai, and Lin will explain the number of genes, gene list, and algorithm when creating ROC in Materials and Methods, and also provide reference numbers in the figure caption.
6 The authors reviewed at least 2 more references on sepsis symptoms, pathology, and diagnosis, such as Reinhart, Konrad, et al. "New approaches to sepsis: molecular diagnostics and biomarkers." Clinical microbiology reviews 25.4 (2012): 609-634. The significance of the criteria for sepsis that the authors focus on should be discussed with referring to those reviews.
7 Referring to Reinhart et al. (2012), there are also diagnostic criteria with an AUC of 0.95. Some of these standards do not use machine learning. I would like to ask the authors to explain in what ways the authors' method is superior or has potential for further development compared to those diagnostic criteria.
8 The main result of this research is a trained machine learning network. Authors should be specific about how they will provide the network and how it will be used by other researchers.
9 The authors have provided the program source code used in this research on Github, but in order to make the source code available to as many researchers as possible, the authors shall prepare a Readme file that explains what data to input and how to input it. An explanation should be added as to how the process proceeds to achieve this goal.
10 The authors state that qRT-PCR of 10 genes in p14 l303-305 is effective for diagnosis. Regarding this conclusion, it is necessary to discuss in what ways this qRT-PCR is superior to the results of serum proteins, etc., which are routinely tested in clinical settings.
11 p15 l370-372: The authors state that "future studies can collect transcriptome data at different time points in sepsis patients and mine pathways and hub genes related to the progression of sepsis over time." In contrast, there is a need to discuss which stage of the progression of sepsis the transcriptional data from this study corresponds to, and how to utilize the changes from this study in future research.
Author Response
Dear Editors and Reviewers:
Thanks for your letter and for the reviewer’s comments concerning our manuscript entitled “Explore the role of different cell death-related genes in sepsis diagnosis based on machine learning algorithm” (ijms-2561747). We have studied comments carefully and have made correction, the answers were listed as follows and the correspondingly revised information marked in red could be found at the revised manuscript. Thanks again.
This study is considered worthy of publication in this journal because it discovered 10 genes that can be used to determine sepsis by reanalyzing the publicly available transcriptome. However, improvements need to be made in the following points.
1.All sepsis authors should discuss the transcriptome material and sepsis information they are referring to. Sepsis has a process from establishment to severity, and it is necessary to discuss the period from the initial stage to severity of the referenced transcriptome by referring to the transcriptome-reported author's papers.
Response: Thanks to the reviewer’s valuable comments. This article used two sepsis related datasets (GSE26440 and GSE95233) from GEO databases, and the clinical information in the datasets does not include the severity of the patient's condition, so we cannot discuss this. In future research, we will collect datasets containing as much clinical information as possible for further discussion.
2 Regarding Figure 3, information on creating the diagram, such as the algorithm, should be added to the caption.
Response: Thanks to the reviewer’s valuable comments. We have supplemented the corresponding analysis method in the title of the subfigure of Figure 3.
3 Regarding Figure 7, an explanation should be added regarding A-Z.
Response: Thanks to the reviewer’s valuable comments. We have supplemented the descriptive information of the subgraphs A-Z in the legend section of Figure 7.
- Explanation should be added regarding A-J in Figure 8 as well.
Response: Thanks to the reviewer’s valuable comments. We have supplemented the description information of subgraphs A-J in the legend section of Figure 8.
5 Regarding Figure 9, Chen, Lai, and Lin will explain the number of genes, gene list, and algorithm when creating ROC in Materials and Methods, and also provide reference numbers in the figure caption.
Response: Thanks to the reviewer’s valuable comments. We have supplemented the gene set information used by Chen, Lai, and Lin and the method information used to construct the diagnostic model in Materials and Methods. The corresponding reference numbers are also supplemented for Figure 9.
- The authors reviewed at least 2 more references on sepsis symptoms, pathology, and diagnosis, such as Reinhart, Konrad, et al. "New approaches to sepsis: molecular diagnostics and biomarkers." Clinical microbiology reviews 25.4 (2012): 609-634. The significance of the criteria for sepsis that the authors focus on should be discussed with referring to those reviews.
Response: Thanks to the reviewer’s valuable comments. We have added relevant content to the introduction section of the revised manuscript and marked it in red.
7 Referring to Reinhart et al. (2012), there are also diagnostic criteria with an AUC of 0.95. Some of these standards do not use machine learning. I would like to ask the authors to explain in what ways the authors' method is superior or has potential for further development compared to those diagnostic criteria.
Response: Thanks to the reviewer’s valuable comments. First, machine learning methods have higher generalizability and accuracy than traditional diagnostic criteria. Because machine learning methods can use large-scale data sets for training, they can also be optimized in specific application scenarios based on strategies such as transfer learning. In contrast, traditional diagnostic standards and methods are usually based on expert experience and small sample studies, and are limited by limited data volume and subjectivity. Therefore, our approach may perform better in clinical settings that deal with large-scale data and variability. Secondly, machine learning methods are more suitable for clinical big data application scenarios, and the trained models can be fine-tuned based on the supply of clinical data. Interpretable clinical features can also be obtained by introducing strategies such as attention mechanisms. This flexibility makes our approach more adaptable to future medical challenges. Finally, we recommend the use of machine learning methods in conjunction with traditional diagnostic criteria in clinical practice. Our model can be used as an auxiliary tool to help doctors make diagnostic decisions more quickly and accurately, improving the overall quality of clinical diagnosis.
8 The main result of this research is a trained machine learning network. Authors should be specific about how they will provide the network and how it will be used by other researchers.
Response: Thanks to the reviewer’s valuable comments. First of all, the machine learning model code and data involved in this article have been open sourced on GitHub. Researchers can obtain the training and usage methods of the model through the README file. Some codes have also been commented for the convenience of researchers.
9 The authors have provided the program source code used in this research on Github, but in order to make the source code available to as many researchers as possible, the authors shall prepare a Readme file that explains what data to input and how to input it. An explanation should be added as to how the process proceeds to achieve this goal.
Response: Thanks to the reviewer’s valuable comments. We have supplemented with the README file on GitHub. The file contains a detailed description of the data and codes for investigator convenience.
10 The authors state that qRT-PCR of 10 genes in p14 l303-305 is effective for diagnosis. Regarding this conclusion, it is necessary to discuss in what ways this qRT-PCR is superior to the results of serum proteins, etc., which are routinely tested in clinical settings.
Response: Thanks to the reviewer’s valuable comments. Generally speaking, PCR is a different detection method that detects target genes at the transcriptional level, rather than proteins. The manuscript does not emphasize the superiority of genetic testing over protein testing, or similar statements. The conclusions obtained in the manuscript were only validated through PCR, and there is still further discussion and repeated validation regarding future clinical applications. In addition, we did not find any relevant expressions at P14 Line 303-305. If there are still inappropriate expressions in the manuscript, please further point out by the reviewer. We are very grateful for this.
11 p15 l370-372: The authors state that "future studies can collect transcriptome data at different time points in sepsis patients and mine pathways and hub genes related to the progression of sepsis over time." In contrast, there is a need to discuss which stage of the progression of sepsis the transcriptional data from this study corresponds to, and
how to utilize the changes from this study in future research.
Response: Thanks to the reviewer’s valuable comments. This paper uses two sepsis datasets (GSE26440 and GSE95233). Among them, GSE26440 contains whole blood-derived RNA samples from 32 sepsis patients 98 hours before admission to the pediatric intensive care unit. The GSE95233 data set contains patients with septic shock who were sampled twice on admission, with the second sample taken on the second or third day. That is, each sample contains data from at least two time points. Therefore, the GSE95233 data set can be applied to mining pathways and core genes related to the progression of sepsis over time. Specifically, time-constrained coefficient learning or a classic deep learning model (suitable for data sets with large sample sizes and many time points) can be applied to model time series data. In addition, attention mechanisms can also be introduced to explore the changes in the importance of disease-related biomarkers over time, providing a reference for drug target discovery. We have added this section to the conclusion

Round 2
Reviewer 4 Report
It is judged that the author's answers and corrections sufficiently address the reviewer's points. However, regarding Figures 7 and 9, the authors shall add the following points as a consideration for readers.
1 Figure7: Please add an explanation of the gray areas above and below the regression line to the figure caption.
2 Figure9: Please add an explanation about the symbols ***, **, * and ns regarding significant differences to the figure caption.
Author Response
Dear Editor and Reviewer,
1 Figure7: Please add an explanation of the gray areas above and below the regression line to the figure caption.
Response: Thanks to the reviewer’s valuable comments. We have added explanations above and below the gray areas in Figure 7 to the text section above Figure 7.
2 Figure9: Please add an explanation about the symbols ***, **, * and ns regarding significant differences to the figure caption.
Response: Thanks to the reviewer’s valuable comments. We added the meaning of ***,* *,* and ns in Figure. 8.